# Prevalence and determinants of unhealthy feeding practices among young children aged 6–23 months in five sub-Saharan African countries

**Berhan Tekeba**[1]*, **Tadesse Tarik Tamir**[1], **Belayneh Shetie Workneh**[2], **Mulugeta Wassie**[3], **Bewuketu Terefe**[4], **Mohammed Seid Ali**[1], **Enyew Getaneh Mekonen**[5], **Alebachew Ferede Zegeye**[6], **Gebreeyesus Abera Zeleke**[5], **Agazhe Aemro**[6]

1 Department of Pediatrics and Child Health Nursing, School of Nursing, College of Medicine and Health Sciences, University of Gondar, Gondar, Ethiopia, 2 Department of Emergency and Critical Care Nursing, School of Nursing, College of Medicine and Health Sciences, University of Gondar, Gondar, Ethiopia, 3 School of Nursing, College of Medicine and Health Sciences, University of Gondar, Gondar, Ethiopia, 4 Department of Community Health Nursing, School of Nursing, College of Medicine and Health Sciences, University of Gondar, Gondar, Ethiopia, 5 Department of Surgical Nursing, School of Nursing, College of Medicine and Health Sciences, University of Gondar, Gondar, Ethiopia, 6 Department of Medical Nursing, School of Nursing, College of Medicine and Health Sciences, University of Gondar, Gondar, Ethiopia

* berishboss7@gmail.com

## Abstract

### Introduction

Despite the World Health Organization's advice against unhealthy feeding, many low- and middle-income countries, including sub-Saharan Africa (SSA) countries, are experiencing a nutritional transition to high in sugar, unhealthy fats, salts, and processed carbohydrates for younger children. However, there is a scarcity of recently updated multicounty information on unhealthy feeding practices and determinants in SSA countries. Therefore, this study aimed to assess the pooled prevalence of unhealthy feeding practices and determinants among children aged 6–23 months in five SSA countries.

### Method

A cross-sectional study design was employed with the most recent demographic and health survey secondary data (DHS) from five SSA countries. This secondary data was accessed from the DHS portal through an online request. The DHS is the global data collection initiative that provides detailed and high-quality data on population demographics, health, and nutrition in low- and middle-income countries. We used a weighted sample of 14,064 children aged 6–23 months. A multilevel mixed-effect binary logistic regression model was fitted to identify significant factors associated with unhealthy feeding practices. The level of statistical significance was declared with p-value < 0.05.

**Data Availability Statement:** Third party data was obtained for this study from the DHS program. Data were requested from at MEASURE DHS program website https://dhsprogram.com/data/available-datasets.cfm. The authors confirm that interested researchers would be able to access these data as the same manner as the authors. The authors also confirm that they had no special access privilege that others wouldn't have.

**Funding:** The author(s) received no specific funding for this work.

**Competing interests:** No competing interset

**Abbreviations:** ANC, Ante natal care; EA, Enumeration Area; ICC, Inter cluster correlation; IYCF, Infant and Young Children Feeding; LR, logistic Regression; MOR, Median Odds ratio; PCV, Proportional Change in Variance; SSA, sub-Saharan Africa; SSB, Sweetened Sugar Beverage; UNICEF, United Nation International Children's Emergency Fund; WHO, World Health Organization.

## Result

This study found that overall, 62.4% (95% CI: 61.62–63.17) of children aged 6–23 months in five SSA countries had unhealthy feeding practices. Rural residents, lower-middle-income SSA countries, and children aged above 12 months had lower odds of unhealthy feeding practices. On the other hand, richer households and women who had not had an optimal antenatal care visit had higher odds of unhealthy feeding practices.

## Conclusion

According to this study, nearly two out of three young children in five SSA countries had unhealthy feeding practices. Both individual and community-level factors are significantly associated with unhealthy feeding practices. As a result, responsible bodies shall make all efforts to reduce unhealthy feeding practices among young children in SSA countries.

## Introduction

The Infant and Young Children Feeding (IYCF) indicator identifies ideal feeding practices for children during their critical growth phase [1]. In a recent update to the IYCF practice, the World Health Organization (WHO) and the United Nations International Children's Emergency Fund (UNICEF) established new indicators, such as consumption of sweetened beverages, consumption of unhealthy/sentinel foods, and zero fruit and vegetable consumption, all of which are collectively referred to as unhealthy feeding practices [2]. These indicators were assessed using a 24-hour dietary recall [3].

Young children between the ages of 6 and 23 months have tremendous potential but also enormous vulnerability. This age group allows for more nourishment in addition to breast milk, which aids the child's physical growth and cognitive development [4]. On the other hand, the first 1000 days of a child's existence are a vital period for developmental delays, deficits, and frequent childhood diseases [4, 5]. Specifically, the introduction of unhealthy food products, such as sugary beverages, sentinel food items, and poor fruit and vegetable consumption, is increasingly practiced in sub-Saharan African (SSA) countries, which has a massive impact on young children [6]. Younger children who consumed unhealthy foods during this period had tremendous negative consequences later in their lives.

The WHO complementary feeding guidelines recommend against providing sugary drinks, like soft drinks, since they just provide energy and may replace more nutrient-dense foods that reduce the consumption of vital vitamins and minerals [7]. Nevertheless, diets in many low-and-middle-income nations including SSA are changing, becoming higher in added sugars, harmful fats, salts, and processed carbs [6, 8, 9]. With rising rates of non-communicable illness and pediatric obesity, the importance of these items in overall diets, as well as their negative influence on nutrition and health, has emerged as a significant worldwide health concern [10]. These diet alterations occur throughout age groups.

Unhealthy young child feeding practices had harmful consequences, including unhealthy weight gain, a higher risk of dental caries, and obesity in childhood [6]. In addition, too much salt in the diet increases the risk of non-communicable disease [11]. According to a previous study, young children who consumed more unhealthy snacks and beverages had lower inadequate micronutrient intake than children who did not consume unhealthy snack food and beverages [6]. Diets deficient in fruits and vegetables also causes children to consume less

nutrients, which can cause non-communicable diseases such as cancer, metabolic diseases like diabetes mellitus, and cardiovascular disease later in life. It can also negatively affect healthy growth and development [6, 12–17].

The global availability of energy-dense, nutrients-poor, and harmful foods and beverages has increased significantly. Young children's purchases of unhealthy meals and sugary-sweetened beverages have reduced in high-income countries [18]. However, expanding urbanization in Africa is associated with rapid change in food surroundings, with possible shifts towards unhealthy food and beverage intake, particularly sentinel foods and sweat beverages [19–21]. Despite the fact that globally fruit and vegetable intake is lower than the recommended amount [22], this burden is significantly higher in low- and middle-income nations, notably in SSA countries [22, 23].

In SSA, the consumption of unhealthy foods and sugary beverages (SSB) is on the rise, largely influenced by economic development and urbanization. This trend aligned with the shift toward a westernized diet [24]. Despite limited data on specific unhealthy foods, prior research showed that high unhealthy food consumption is in particular significant in SSA due to affordability, accessibility, and aggressive marketing by the beverage industry [25]. This dietary change poses challenges to the public health system. Overall, addressing sugar-sweetened beverage and sentinel food consumption, including chips, chocolate, sweat, and candies, among children in SSA requires strong policies, better nutritional education, and regulatory measures [11, 26]. In addition, fruit and vegetable consumption in SSA is generally below the global recommendation related to socio-economic challenges, poor access to fresh fruit and vegetables, inadequate storage facilities, and seasonal variability [27].

To ensure optimum growth, development, and health during the early years of life, healthy feeding practice is recommended [28]. Despite this, limited resources, cultural beliefs, the maternal education gap, and sole consumption of monotonous starchy diets contribute to unhealthy feeding practices in SSA. However, as far as our level of knowledge is concerned, there is a paucity of current multicounty level information on the magnitude and determinates of unhealthy feeding practices, including sweat beverages, sentinel/sweat beverage, and zero fruit or vegetable consumption among children aged 6–23 months in resource-limited SSA countries. Therefore, this study aimed at assessing the pooled prevalence of unhealthy feeding practices and determinants among children aged 6–23 months in five SSA countries, which helps inform concerned bodies to make appropriate interventions accordingly.

## Methods

### Study setting, design and period

We determined the pooled prevalence and associated factor of unhealthy feeding practices among five SSA nations using the most recent data from the Demographic and Health Survey (DHS). Since our outcome was a new nutritional indicator that requires the most recent DHS, we only included countries that had the most recent DHS (2021–2023). The countries selected were Burkina Faso, Ghana, Kenya, Mozambique, and Tanzania. Every five years, the DHS surveys are routinely conducted in low- and middle-income countries using standardized, pretested, and validated questionnaires [29]. In order to enable multi-country analysis, the survey adheres to a comparable standard approach for the creation of questions, sampling, data collection, and coding.

### Data source and sampling procedure

Our data source was the MEASURE DHS program. The DHS Program is a global data gathering project that offers detailed and high-quality information about population demographics,

health, and nutrition. DHS surveys, mostly funded by the USAID, are undertaken in low- and middle-income countries around the world to help governments, researchers, and organizations monitor and improve public health policies and social services.

The DHS surveys use a stratified, two-stage cluster sampling technique. The first step was the random selection of clusters, or enumeration areas (EAs), that encompass the entire country from the sampling frame derived from the most recent national survey that was made accessible. EAs are small, well-defined geographic areas often delineated as a part of a national census that serve as a building block for the national survey, which is used as a sampling unit. Each EA is designed to include a manageable number of households. In the second step, interviews were held in a subset of the target population's households using systematic sampling, which was applied to all of the households mentioned in each cluster (women aged 15–49). A weighted sample of 14,604 child-mother pairs within the five years prior to the survey in each country was included in this study. For households that had more than one child in the five years before the survey, the youngest child was taken into account. However, as with a large-scale survey, there were some exclusion, including hard-to-reach populations such as remote areas and conflict areas, and transient populations were not included. Additionally, a missing value for the outcome variable was not included in the study (n = 54).

## Study and source population

The source population consisted of all infants and young children aged 6–23 months in five SSA countries. The study population consisted of infants and young children aged 6–23 months in the five years preceding the survey period in the selected EAs in the respective countries. Enumeration is the primary sampling unit of the survey clusters.

## Inclusion criteria and exclusion criteria

Infants and young children aged 6–23 months born in countries that had a DHS report since 2022 in SSA were included. SSA countries that had not reported DHS in the specified period of time (since 2021) were excluded from the study. Additionally, a missing value for the outcome variable was excluded from the study.

## Outcome variable

The study outcome variable was unhealthy feeding practice. Unhealthy feeding practices include sweet beverage consumption, unhealthy (sentinel) food consumption, and zero-fruit and vegetable consumption. Then, the unhealthy feeding practice was determined by combining the above indicators. Each indicator had food items to be consumed. Sweet beverages include chocolate-flavored drinks, sodas, malts, sports drinks, and super energy drinks. Unhealthy food/sentinel consumption items include chocolate, sweets, candies, pastries, chips, and Crips. Fruit and vegetable items include pumpkin, carrot, green leafy vegetables, vitamin A-rich foods, and other fruit and vegetable items. Then, a child who consumes each food item in each category is considered to have consumed a sweet beverage or unhealthy food. Zero fruit and vegetable consumption was coded "yes" for those who didn't consume fruit and vegetable items and "no" for those who consumed fruit and vegetable items. Finally, unhealthy feeding practices were calculated from the sum of sweet beverage consumption, sentinel food consumption, and zero fruit and vegetable consumption. Those children who consumed the above two indicators and who didn't consume fruit and vegetables were considered to have unhealthy feeding practices based on 24 hour dietary recall.

### Operational definitions

**Unhealthy feeding practice.** The percentage of children who feed sweaty beverages, sentinel unhealthy foods, and did not feed any fruit or vegetables in the previous day [7].

**Sweat beverage consumption.** Children who consume sweat beverages, including juice, herbal drink, tinned or powdered milk, sweetened liquid, soda, malt, and chocolate-flavored drink in the previous day [7].

**Sentinel unhealthy food.** Children who consume chocolate, sweats, candies, pastries, chips, crisps, French fries, fried dough, and instant noodles in the previous day [7].

**Zero-fruit or vegetable consumption.** A child who didn't consume any fruit or vegetables in the previous day, including pumpkin, carrot, squash, dark green leafy vegetables, mango, papaya, and any other fruit [7].

### Independent variables

Both individual and community-level factors were reviewed from different literatures, and these include maternal age, child sex, child age, maternal educational level, media exposure, place of delivery, antenatal care (ANC), mother-work status, women's involvement in household decision-making, and household wealth quintiles. residence, distance to a health facility, community ANC utilization, community women's education, community media exposure, and community wealth status were community-level factors aggregated from individual-level factors [30–34].

Wealth index (WI) were categorized as poor, middle, and rich [35]. ANC use was classified as optimal if a mother had more than 8 visits during her pregnancy and non-optimal if the mother had less than 8 ANC visits during her pregnancy [36]. Maternal education status was categorized as no formal education, primary, secondary, and above. Media exposure was aggregated from three variables in the DHS data set. These variables are: 1) frequency of reading a newspaper or magazine; 2) frequency of listening to radio; and 3) frequency of watching television. These three variables each have three responses. a. not at all; b. less than once a week; c. at least once a week. Then, we merged the above three variables as one variable, and we categorized not at all responses as "having no media exposure" and less than once and at least once a week as "having media exposure" [37].

### Community level variables

Community-level variables were countries economy level, community illiteracy, community antenatal care use, community media exposure, and community-level poverty. Countries economic levels were categorized as "low income" and "lower middle income" based on the World Bank countries growth report. Low-income countries include Burkina Faso and Mozambique; lower-middle-income countries include Ghana, Kenya, and Tanzania.

Community illiteracy, community ANC use, community media exposure, and community-level poverty were aggregated from individual women's characteristics of education, ANC use media exposure, and wealth index, respectively. After aggregation, community level variables were categorized as high and low based on the median value. Aggregation was done by using Excel and Stata software.

### Data management and model selection

After being extracted from the DHS portal, Stata version 14 was used to enter, code, clean, record, and analyze the data. In DHS, data variables are nested by clusters, and those within the same cluster show more similarities than those with separate clusters. Thus, using the traditional logistic regression model violates the assumptions of independent observation and

equal variance across clusters. Therefore, a multi-level logistic regression analysis was employed in this study in order to account for the hierarchical nature of DHS data.

## Fixed effect analysis

A bivariate multi-level logistic regression model was employed in the study to identify the variables associated with unhealthy feeding practice. In the analysis, four models were fitted. The first (null) model contains only the outcome variables to test random variability and estimate the intra-cluster correlation coefficient (ICC). The second model contains individual-level variables; the third model contains only community-level variables; and the fourth model (model III) contains both individual-level and community-level variables [38]. A p-value of 0.05 was used to define statistical significance. Adjusted odds ratios (AOR) with corresponding 95% confidence intervals (CIs) were calculated to identify independent predictors of unhealthy feeding practices.

For the bivariate and multivariate multilevel logistic regression analyses, the Stata syntax xtmelogit was used [39].

$$logit(\pi ij) = log[\pi ij/(1 - \pi ij)] = \beta0 + \beta1xij + \beta2xij \ldots + u0j + e0ij,$$

Where, $\pi ij$: the probability of the ith unhealthy feeding practice, $1 - \pi ij$: the probability of not practicing unhealthy feeding, $\beta0$: intercept, $\beta n$: regression coefficient Xij: independent variables u0j: community level error, e0ij: individual level errors [40].

## Random effect analysis

Variation of the outcome variable or Random effects were assessed using the proportional change in variance (PCV), intra-class correlation coefficient (ICC), and median odds ratio (MOR) [41, 42].

The ICC shows the variation in unhealthy feeding practice due to community characteristics which was calculated as: ICC = Vnull/ (Vc + $\pi^2/3$), where, Vnull, variance of null model, Vc, variance of the cluster [43]. The higher the ICC, the more relevant the community characteristics for understanding individual variation in unhealthy feeding practice.

MOR is the median value of the odds ratio between the areas with the highest unhealthy feeding practice and the area with the lowest unhealthy feeding practice when randomly picking out two neonates from two clusters, which was calculated as: MOR = $exp(\sqrt{}*Vc*0.6745) \sim MOR = exp(0.95*vc)$, where Vc is the variance of the cluster.

Furthermore, the PCV illustrates how different factors account for variations in the prevalence of unhealthy feeding practice and is computed as $PCV = (Vnull - Vc)/Vnull$, where Vc is the cluster-level variance and Vnull, is the variance of the null model [44, 45]. The likelihood of an unhealthy feeding practice and independent variables at the individual and community levels were estimated using both fixed effects and random effect analysis. An AOR and 95% confidence intervals with a p-value of 0.05 were used to assess the strength of association. Due to the hierarchical nature of the model, models were compared using deviation = -2 (log likelihood ratio), and the best-fit model was determined by taking the model with the lowest deviance [46–48]. By calculating the variance inflation factors (VIF), the variables employed in the models were checked for multi-collinearity; the mean value of the VIF of the final model was 1.73.

## Ethical consideration

The authors analyzed secondary, publicly available data obtained from the DHS program database. There was no additional ethical approval, and informed consent was obtained by the authors. In order to perform our study, we registered with the DHS web archive, requested the dataset, and were granted permission to access and download the data files. According to the

**Table 1. Country, sample size, survey year of studies and prevalence of unhealthy feeding practice in five SSA countries.**

| County | Sample size | Prevalence | Survey year |
|---|---|---|---|
| Burkina Faso | 3,419 | 56.52% | 2021 |
| Ghana | 2,680 | 65.99% | 2022 |
| Kenya | 2,502 | 58.96% | 2022 |
| Mozambique | 2,756 | 58.5% | 2022/23 |
| Tanzania | 3,247 | 66.55% | 2022 |
| Total | 14,604 | 61.28% | 2021/2023 |

DHS report, all participant data was anonymized during the collection of the survey data. More details regarding DHS data and ethical standards are available online at https://dhsprogram.com/data/.

## Result

Our study comprised a weighted sample of 14,604 young children aged 6 to 23 from five SSA countries. The highest prevalence was found in Tanzania (66.55%), and the lowest prevalence was observed in Burkina Faso (56.52%) **Table 1**.

### Socio-demographic, child, maternal, and health care utilization characteristics

A total of 14,064 children aged 6–23 months were included in the study. The mean age of mothers was 28.27±0.05 years; more than two-thirds (71.14%) of mothers fall in the age range of 20–34 years. More than two-thirds (68.32%) of mothers completed primary education. More than half of mothers (54.98%) were at work, and nearly two-thirds (61.3%) of mothers were involved in the household health care decision-making process. The majority (90.9%) of mothers hadn't optimal ANC (had less than 8 visits), and more than two-thirds (70.02%) of women's had no postnatal visits by skilled professionals. The mean age of children's was 14 ± 0.05 months. More than two-thirds (67.95%) of children reside in rural areas. More than half (51.68%) of children were from highly illiterate communities. More than half (50.94%) of the children were also from low media-exposed communities **Table 2**.

### Prevalence of unhealthy feeding practice (sweet beverage consumption, unhealthy/sentinel food consumption, zero-fruit and vegetable consumption)

The overall pooled prevalence of unhealthy feeding practices in five SSA countries was 62.4% (95% CI: 61.62–63.17) **Fig 1**.

The prevalence of sweet beverage consumption in five SSA African countries was 15.37% (95% CI: 14.8–15.94). The lowest prevalence was observed in Kenya (10.73%), and the highest was observed in Ghana (22.20%). The prevalence of unhealthy/sentinel food consumption was 17.66% (17.05–18.27). The lowest prevalence was observed in Tanzania (11.32%), and the highest prevalence was in Ghana (23.40%). Regarding zero fruit and vegetable consumption, the overall prevalence in five African countries was 40.65 (95% CI: 39.87–41.44). The highest prevalence was held by Tanzania (52.16%), and the lowest prevalence was in Ghana (33.34%) **Figs 2–4**.

### Random effect and model fit statistics

As shown in Table 3, the ICC in the null model was 3.34, indicating that 3.34% of the variations in unhealthy feeding practices were attributed to cluster differences. The MOR value of

**Table 2. Socio-demographic and community level characteristics of study participants.**

| Variables | Response | Frequency | Percentage |
|---|---|---|---|
| **Socio-demographic characteristics** | | | |
| Maternal education | Un-educated | 4,627 | 31.68 |
| | Primary | 4,914 | 33.65 |
| | Secondary & above | 5,063 | 34.67 |
| Maternal age | < 20 years | 1,316 | 9.01 |
| | 20–34 years | 10,389 | 71.14 |
| | ≥35 years | 2,899 | 19.85 |
| Marital status | Married | 8,765 | 60.02 |
| | Other | 5,839 | 39.98 |
| Husband education | Un-educated | 4,361 | 34.64 |
| | Primary | 3,804 | 30.21 |
| | Secondary | 4,426 | 35.15 |
| House hold head sex | Male | 11,431 | 78.27 |
| | Female | 3,173 | 21.73 |
| Mother currently working | No | 6,574 | 45.02 |
| | Yes | 8,030 | 54.98 |
| Number of under-five | ≤2 | 11,664 | 79.87 |
| | ≥3 | 2,940 | 20.13 |
| Media exposure | No | 4,821 | 33.01 |
| | Yes | 9,783 | 66.99 |
| Child sex | Male | 7,374 | 50.49 |
| | Female | 7,230 | 49.51 |
| Currently breast feeding | No | 2,927 | 20.04 |
| | Yes | 11,677 | 79.96 |
| Birth order | First | 3,567 | 24.43 |
| | Second-third | 3,638 | 38.67 |
| | Fourth & above | 5,399 | 36.97 |
| Post natal check | No | 9,817 | 70.02 |
| | Yes | 4,203 | 29.98 |
| ANC check | Non-optimal | 12,746 | 90.9 |
| | Optimal | 1,275 | 9.1 |
| Wealth index | Poor | 6,345 | 43.45 |
| | Middle | 2,854 | 19.54 |
| | Rich | 5,405 | 37.01 |
| Women involved in decision making | Yes | 7,730 | 61.3 |
| | No | 4,881 | 38.7 |
| **Community level factors** | | | |
| Place of residence | Urban | 4,681 | 32.05 |
| | Rural | 9,923 | 67.95 |
| Economic level | Low income | 6,004 | 41.11 |
| | Lower middle income | 8,600 | 58.89 |
| Community illiteracy | High | 7,491 | 51.68 |
| | Low | 7,006 | 48.32 |
| Community ANC use | Low | 8,573 | 58.74 |
| | High | 6,021 | 41.26 |
| Community media exposure | Low | 7,439 | 50.94 |
| | High | 7,165 | 49.06 |

(*Continued*)

**Table 2.** (Continued)

| Variables | Response | Frequency | Percentage |
|---|---|---|---|
| Community poverty level | High | 7,214 | 49.4 |
| | Low | 7,390 | 50.6 |

**ANC:** Antenatal care

1.38 in the null model also depicted that the odds of unhealthy feeding practice among study subjects were different between high and low clusters. Furthermore, the PCV revealed that 25.7% of the variation in unhealthy feeding practice among study subjects was explained by the final model with the lowest deviance, which was the better-fitted model, which was model III, and was selected for interpretation by the final model (model III). Regarding model comparison and fitness, deviance was used. The model with the lowest deviance was the better-fitted model, which was model III, and was selected for interpretation **Table 3**.

## Factors associated with un-healthy feeding practice in five SSA countries

In the final model (model III) of the multivariable multilevel logistic regression model, child age, non-optimal ANC check, rural residence, rich wealth index, lower middle-income economic level, and region were significantly associated with unhealthy feeding practices.

Children aged 12–23 months had 38% reduced odds (AOR = 0.62; 95% CI: 0.58–0.68) of unhealthy feeding practice as compared to children aged 6–11 months. Children from women having non-optimal ANC visits had 69% increased odds (AOR = 1.31; 95% CI: 1.1–1.38) of unhealthy feeding practices as compared to children from women having optimal ANC visits. Young children from rich households had 77% increased odds (AOR = 1.23; 95% CI: 1.1–1.38) of unhealthy feeding practices as compared to children from poor wealth quintiles. Children residing in rural areas had 21% reduced odds (AOR = 0.79; 95% CI: 0.71–0.88) of unhealthy feeding practices as compared to children residing in urban areas. Children from lower-

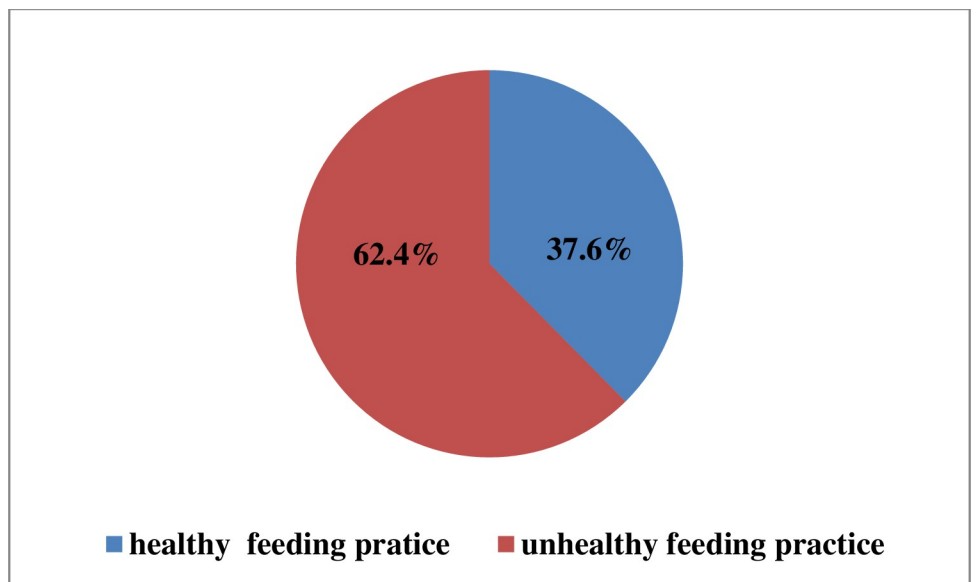

**Fig 1. The overall pooled prevalence of unhealthy feeding practice among children aged 6–23 months in five SSA countries.**

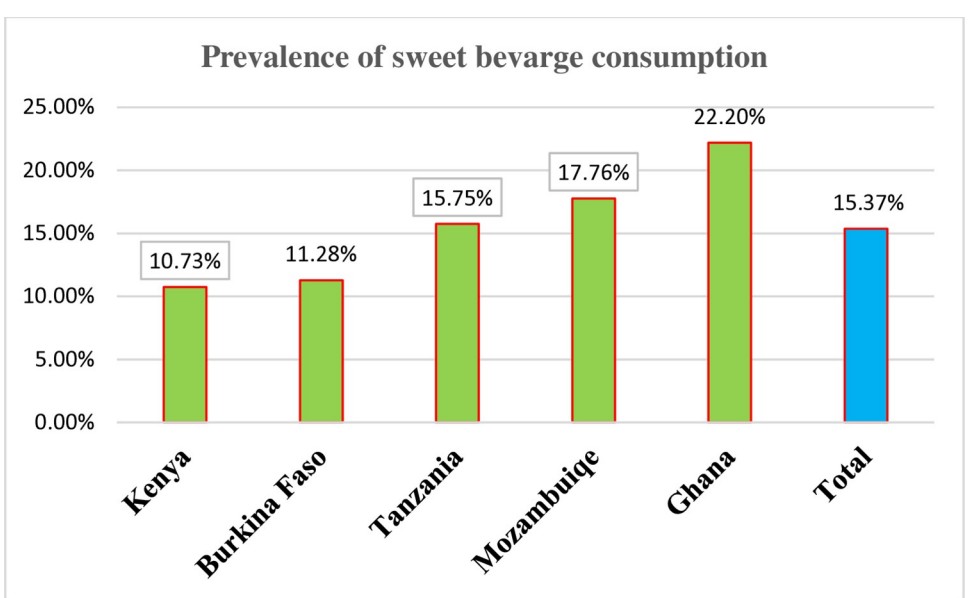

**Fig 2. The prevalence of sweet beverage consumption among children aged 6–23 months in five SSA countries.**

middle-income countries had 24% reduced odds of (AOR = 0.76; 95% CI: 0.69–0.84) unhealthy feeding practices as compared to low-income countries **Table 4**.

## Discussion

Many low- and middle-income countries' diets are trending toward increased consumption of added sugars, unhealthy fats, salts, and processed carbs. Commercially prepared foods are frequently calorically packed, nutritionally deficient, and rich in salt, sugar, saturated or trans fatty acid [3]. Various guiding documents and the WHO recommend the need to avoid or

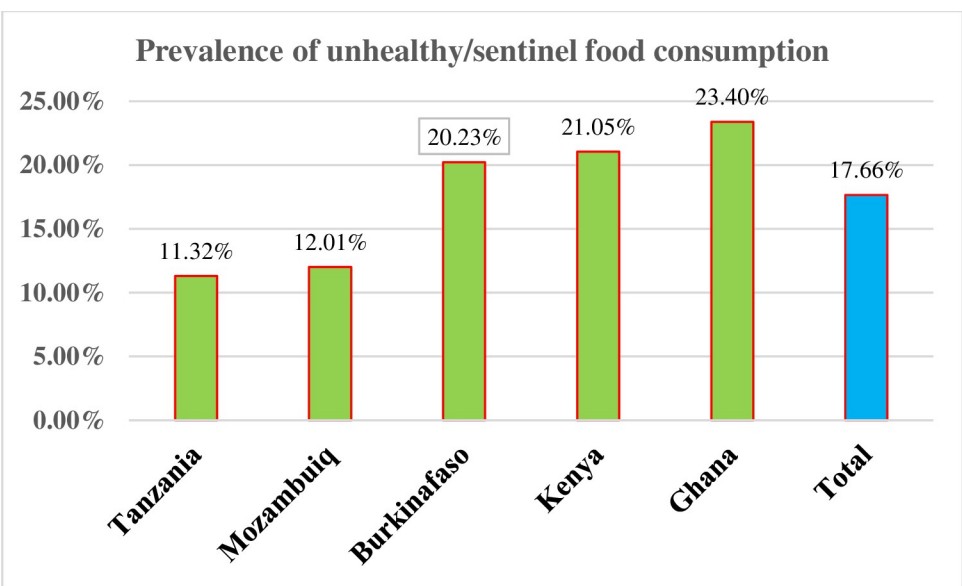

**Fig 3. The prevalence of unhealthy/sentinel food consumption among children aged 6–23 in five SSA countries.**

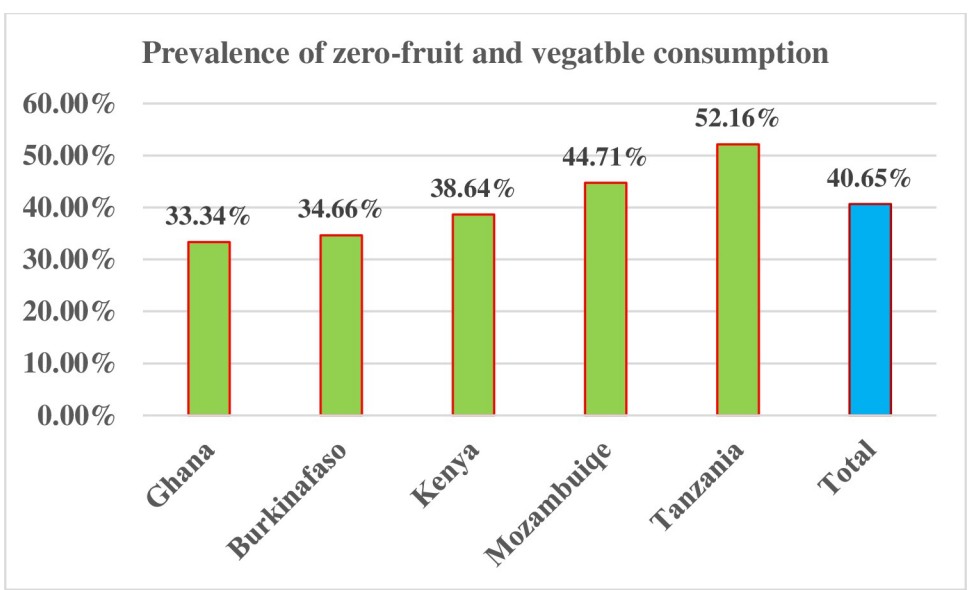

**Fig 4. The prevalence of zero fruit and vegetable consumption among children aged 6–23 months in five SSA countries.**

limit these types of foods when feeding infant and young children; thus, estimating the pooled prevalence and identifying potential predictors of unhealthy feeding practices is a key step towards improving healthy feeding practices in young children.

In this study, the pooled prevalence of unhealthy feeding practices in five SSA nations was 62.4% (95% CI: 61.62–63.17). This study finding is lower than the study done in United States of America (USA) (84.4%), Argentina (90.8%), and Brazil (94%) [49–51]. The possible explanation could be that even though consumption of unhealthy foods, including sentinel foods, declines in high-income nations, a higher number of young children still consume these sugar-added foods due to economic ability to purchase processed foods and sweat beverages [18]. In addition, sweat beverages and processed foods are widely available, affordable, and heavily marketed in the USA. A combination of widespread marketing, economic factors, convenience, lifestyle, and urbanization in the USA, Brazil and Argentina leads to a higher prevalence of unhealthy feeding practices than in SSA [2].

**Table 3. Random effect and model fit statistics of unhealthy feeding practice among children aged 6–23 months in five SSA countries.**

| Parameter | Null model | Model I | Model II | Model III |
|---|---|---|---|---|
| **Variance** | 0.11 | 0.09 | 0.1 | .08 |
| **ICC** | 3.34 | 2.57 | 3.8 | 2.5 |
| **MOR** | 1.38 | 1.32 | 1.34 | 1.32 |
| **PCV** | Reference | 23.69% | 16.51 | 25.7% |
| **Model fitness** | | | | |
| **LLR** | -9997.17 | -8216.94 | -9853.55 | -8192.44 |
| **Deviance** | 19,994.34 | 16,433.88 | 19,707.1 | 16,384.88 |

**ICC:** Intra cluster correlation coefficient, **MOR:** Median odds ratio

**PCV:** Proportional change in variance, **LLR:** Log-likely-hood ratio

**Table 4. Individual and community level factors associated with unhealthy feeding practice among children aged 6–23 months in five SSA countries.**

| Individual and community level factors | Response | Model I | Model II | Model III |
|---|---|---|---|---|
| | | AOR (95% Cl) | AOR (95% Cl) | AOR (95% Cl) |
| Maternal education | Un-educated | 1 | | 1 |
| | Primary | 1.0 (0.91–1.11) | | 0.96 (0.86–1.06) |
| | Secondary & above | 1.12 (0.99–1.27) | | 1.11 (0.98–1.26) |
| Maternal age | <20 years | 0.95 (0.81–1.11) | | 0.92 (0.79–1.09) |
| | 20–34 years | 1 | | 1 |
| | ≥35 years | 1.0 (0.90–1.10) | | 0.99 (0.89–1.10) |
| Marital status | Married | 0.93 (0.85–1.01) | | 0.98 (0.90–1.07) |
| | Other | 1 | | 1 |
| Husband education | Un-educated | 0.89 (0.79–1.00) | | 0.91 (0.81–1.03) |
| | Primary | 1.03 (0.92–1.15) | | 0.99 (0.89–1.11) |
| | Secondary | 1 | | 1 |
| House hold head sex | Male | 1.04 (0.96–1.11) | | 0.95 (0.86–1.06) |
| | Female | 1 | | 1 |
| Mother currently working | No | 1 | | 1 |
| | Yes | 0.95 (0.88–1.02) | | 0.97 (0.90–1.05) |
| Number of under-five | ≤2 | 1 | | 1 |
| | ≥3 | 0.97 (0.88–1.06) | | 0.97 (0.8–1.07) |
| Media exposure | No | 1 | | 1 |
| | Yes | 1.00 (0.92–1.09) | | 1.05 (0.96–1.15) |
| Child sex | Male | 1 | | 1 |
| | Female | 0.62 (0.57–0.68) | | 1.04 (0.96–1.12) |
| Child age | 6–11 | 1 | | 1 |
| | 12–23 | 0.62(0.57–0.68) | | 0.62 (0.58–0.68)* |
| Currently breast feeding | No | 1 | | 1 |
| | Yes | 0.92 (0.83–1.01) | | 0.94 (0.85–1.04) |
| Birth order | First | 0.95 (0.83–1.08) | | 0.95 (0.84–1.08) |
| | Second-third | 0.94 (0.85–1.03) | | 1 |
| | Fourth & above | 1 | | 0.94 (0.86–1.04) |
| Post natal check | No | 1 | | 1 |
| | Yes | 0.94 (0.87–1.02) | | 0.97 (0.89–1.05) |
| ANC check | Optimal | 1 | | 1 |
| | Non-optimal | 1.29 (1.12–1.48) | | 1.31(1.13–1.52)* |
| Wealth index | Poor | 1 | | 1 |
| | Middle | 1.09 (0.99–1.21) | | 1.04 (0.94–1.16) |
| | Rich | 1.43 (1.29–1.57) | | 1.23 (1.10–1.38)* |
| Women involved in decision making | Yes | 0.98 (0.90–1.06) | | 0.95 (0.88–1.03) |
| | No | 1 | | 1 |
| **Community level factors** | | | | |
| Place of residence | Urban | | 1 | 1 |
| | Rural | | 0.69 (0.6–0.75) | 0.79 (0.71–0.88)* |
| Economic level | Low income | | 1 | 1 |
| | Lower middle | | 0.78 (0.72–0.8) | 0.76 (0.69–0.84)* |
| Community illiteracy | High | | 1 | 1 |
| | Low | | 1.04 (0.95–1.14) | 1.05 (0.95–1.15) |
| Community ANC use | Low | | 1 | 1 |
| | High | | 1.06 (0.97–1.15 | 1.00 (0.91–1.09) |

*(Continued)*

**Table 4.** (Continued)

| Individual and community level factors | Response | Model I | Model II | Model III |
|---|---|---|---|---|
| | | AOR (95% Cl) | AOR (95% Cl) | AOR (95% Cl) |
| Community media exposure | Low | | 1 | 1 |
| | High | | 1.00 (0.91–1.09) | 0.98 (0.89–1.09) |
| Community poverty level | High | | 1 | 1 |
| | Low | | 0.94 (0.86–1.03) | 0.99 (0.89–1.09) |

\*, statistically significant (p-value<0.05), **AOR**: Adjusted odds ratio

**ANC**: Antenatal care, **CI:** Confidence interval

The prevalence of unhealthy/sentinel food consumption in this study was 17.66% (17.05–18.27). This is lower than the study done in Ethiopia (63.7%) [33] and Brazil (43.1%) [52]. The possible explanation could be a difference in sample size, study period, study area, and socio-economic status. Thus, previous studies were conducted in single nations, but our study was a pooled estimate of five countries, resulting in a large sample size and increased representativeness. In addition, urbanization in Brazil had widespread access to processed foods. But our finding is higher than the previous SSA study (13.41%). The possible explanation could be that we used the most recent DHS Guide-8 indicator to identify food items to be consumed to assess the indicator, whereas prior studies used a non-updated 2021 WHO indicator for assessing infant and young children's feeding practices [32]. Furthermore, different SSA countries were used to assess the unhealthy feeding practices. Furthermore, rapid urbanization in SSA has led to major lifestyle shifts; marketing and advertising targeted young children; westernization of diets; the influence of the global food industry; and a lack of public health awareness are responsible for this increasing trend [53].

The prevalence of sweet beverage consumption in five SSA countries was 15.37% (95% CI: 14.8–15.94), which is lower than the study done in four African and Asian countries [54] and Cambodia (57%) [55]. This might be due to differences in the study period, economic level, and study settings; thus, these studies were done prior to our study, and nutritional recommendations and guidelines were updated. Moreover, Cambodia had a better economy than most African nations, which was accompanied by a transition away from traditional dietary patterns towards westernized diets, including sweet beverages [8]. In addition, prior studies were done in urban areas, which further increased the estimate of sweet beverage consumption since unhealthy food consumption is higher among urban residents than rural residents [56].

Regarding zero fruit and vegetable consumption, the overall prevalence in five SSA countries was 40.65% (95% CI: 39.87–41.44).The prevalence of zero-fruit and vegetable consumption was lower than the study done in low- and middle-income countries (45%) [30]. The possible explanation could be the difference in study period between low and middle-income nations, which was done by using the DHS from 2006–2020, but we used the most recent DHS data set from 2021–2023 that countries might be aware of for healthy feeding, including increasing energy-dense and nutrient-rich foods like fruit and vegetables. In addition, we used a small number of countries, whereas prior studies used 64 low- and middle-income countries, which might be a true and convenient estimate.

Children aged 12–23 months were 38% less likely to consume unhealthy food as compared to children aged 6–11 months. This is supported by other studies [33]. This might be due to the fact that many parents introduce complementary foods to their newborns between the ages of 6 and 11 months, but occasionally they rely on prepackaged baby foods, fruit juices, and sweetened purees because they believe these to be suitable for young infants. In addition,

children aged 12–23 months have a higher probability of eating a family diet, which decreases unhealthy food consumption [57]. But this finding is contrary to other studies done in different parts of the world [54, 58]. The possible explanation could be children aged 6–11 months preference and strong demand for sweet and convenient foods to begin complimentary feeding at a young age.

Children from wealthier households were 23% more likely to practice unhealthy feeding as compared to children from poor wealth status. This is supported by other studies [32, 59]. The possible explanation could be that richer households, due to their higher income, might buy precious and luxurious commercially prepared food items for their children to show up or may have the wrong perception, like giving sweet beverages more nutritional value for young children's. In contrast, poor families rely on other cheap foods like high-fiber diets. In summary, richer households are more likely to consume unhealthy, commercially processed foods due to greater financial means, access, timing constraints, dining out, and exposure to marketing, as well as lifestyle factors that favor convenience over nutrition [60]. Therefore, informing richer households to reduce unhealthy feeding practices by responsible bodies is crucial to reducing the long-term side effects of these inappropriate dietary patterns.

Children from rural communities were 26% less likely to practice unhealthy feeding as compared to urban children. This is supported by other studies [6, 33, 61]. One explanation for this could be that moms in urban locations may be able to obtain and buy various types of unhealthy food for their children from small businesses, the streets, and supermarkets. Women who live in cities may experience food insecurity and decide to buy cheaper food from the market [62, 63]. The metropolitan populace is another potential demographic, as they frequently choose unhealthy foods that are readily available and affordable [56]. Conversely, children living in rural households eat a diet rich in fiber that consists of roots, cereals, and grains that are obtained directly from agricultural products. Foods high in nutrients, such as fruits, vegetables, and animal products, are rarely consumed, particularly among young children [64]. Furthermore, limited access to commercially processed food, homegrown traditional food consumption, cultural practice, and economic constraints make rural children less likely to consume unhealthy sweat beverages and processed groceries.

Mothers with no optimal ANC visit were 69% more likely to practice unhealthy child-feeding practices as compared to those with an optimal ANC visit. This is supported by other studies [16]. This could be explained by the fact that, in contrast to subpar ANC visits, optimal ANC visits offer a chance to improve mother and child nutrition in low- and middle-income nations. For example, during ANC, healthy eating is stressed, emphasizing the value of fruits and vegetables as well as low-sugar and low-salt foods. Mothers who don't have optimal ANC visits are at high risk of practicing unhealthy feeding later to their young child due to lack of education and information, poor confidence in child feeding secondary to inadequate follow-up, and poor support regarding health feeding practice. Hence, it is essential to strengthen and optimize antenatal care services during pregnancy to have safe delivery and inform healthy feeding later in child life.

Lower-income countries had higher odds of unhealthy child feeding practices as compared to lower-middle-income countries. This study is supported by a Brazilian study [52, 65, 66]. This might be even though an improved economy results in higher expenditure and intake of extra calories, such as sweet beverages, but the affordability of these food items, legislation against marketing these processed food items, and cultural influence might positively influence the consumption of unhealthy foods in low-income African countries [67]. Furthermore, low utilization of fruit and vegetables was higher in low-income countries [16], by which countries with the lowest economy couldn't afford to buy a diversified diet including fruit and vegetables for their children, which might further strengthen our analysis. We can conclude that

unhealthy feeding practices can be plasticized both in low-and higher-income countries. Economic independence alone is not sufficient to explain unhealthy feeding practices; additional factors, including countries policies and regulations to produce sweat beverages, individual preferences for food items, cultural influence, food marketing and advertising, and level of education and information on unhealthy foods, matter to the practice of unhealthy feeding in young children. Therefore, context-specific solutions and interventions aimed at improving access to healthy foods, raising awareness about proper nutrition, and regulating the marketing of unhealthy food for these vulnerable young children should be the responsibility of executors and every individual.

## Strength

The strength of this study was a multi-country survey with weighted pooled national sample data; thus, it has the potential to help programmers and policymakers develop effective interventions at the multi-country level.

## Limitation

Due to the cross-sectional nature of the data, it cannot establish a cause-and-effect relationship, and the DHS is primarily dependent on respondents' self-reports, so there is a chance of recall bias on food items a child consumes in 24 hours in our study. In addition, in this study, we included only a few countries from SSA, which leads to underrepresentation. Moreover, some food items were missed in some countries DHS data, so there might be overestimation or underestimation.

## Research and policy implication

This study has policy implications for improving infant and child feeding practices. Based on the high magnitude of the problem, governments of these African countries should give special attention to infant and young children's feeding practices by advising the family, community, and countries to reduce sweat beverage consumption, unhealthy or sentinel food consumption, and improve fruit or vegetable consumption. In addition, based on identified factors, health care providers and the ministries of health of respective countries should promote optimal ANC visits, reduction of starch and salt foods for very young children (<1 year) in dishes, and rich communities should also reduce sweat beverages and unhealthy or sentinel food consumption.

Furthermore, policymakers and programmers should also revise and follow strict enforcement laws against the distribution, usage, and licensure of commercially prepared food items, particularly for youngsters.

Future research incorporating more countries to strengthen generalizability shall be done.

## Conclusion

According to this study, nearly two out of three young children (62.4%) in five SSA countries, namely Burkina Faso, Ghana, Kenya, Mozambique, and Tanzania, had unhealthy feeding practices. This indicates it is a significant public health issue in SSA countries. Both individual and community-level factors are significantly associated with unhealthy feeding practices. Unhealthy feeding practices during early childhood lead to obesity, increased risk of diabetes, cardiovascular disease, and cancer later in life. To mitigate these risks, immediate action is required, including policies to limit marketing unhealthy food to children and strategies to improve access to healthy foods. Policymakers and other relevant authorities shall give special

attention to young infant feeding practices in richer households, urban children, young infants, women with no antenatal care use, and low-income African countries. Therefore, awareness creation for rich households and urban younger children to reduce unhealthy feeding, improving antenatal care utilization among women, special attention to very younger children, and bringing long-term improvement in countries' economies are essential to tackling unhealthy feeding practices in these SSA countries.

## Author Contributions

**Conceptualization:** Berhan Tekeba, Tadesse Tarik Tamir, Bewuketu Terefe, Mohammed Seid Ali, Enyew Getaneh Mekonen, Alebachew Ferede Zegeye, Gebreeyesus Abera Zeleke, Agazhe Aemro.

**Data curation:** Berhan Tekeba, Belayneh Shetie Workneh.

**Formal analysis:** Berhan Tekeba, Mulugeta Wassie, Mohammed Seid Ali.

**Investigation:** Tadesse Tarik Tamir.

**Methodology:** Berhan Tekeba, Mulugeta Wassie, Bewuketu Terefe, Alebachew Ferede Zegeye, Gebreeyesus Abera Zeleke, Agazhe Aemro.

**Software:** Enyew Getaneh Mekonen, Alebachew Ferede Zegeye.

**Supervision:** Tadesse Tarik Tamir, Mohammed Seid Ali, Agazhe Aemro.

**Validation:** Belayneh Shetie Workneh, Bewuketu Terefe, Enyew Getaneh Mekonen, Gebreeyesus Abera Zeleke.

**Visualization:** Belayneh Shetie Workneh.

**Writing – original draft:** Berhan Tekeba, Belayneh Shetie Workneh, Mulugeta Wassie, Bewuketu Terefe, Mohammed Seid Ali, Enyew Getaneh Mekonen, Alebachew Ferede Zegeye, Agazhe Aemro.

**Writing – review & editing:** Berhan Tekeba, Tadesse Tarik Tamir, Gebreeyesus Abera Zeleke, Agazhe Aemro.

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
