## [Decision Letter · Decision Letter 0]

11 Nov 2024

PONE-D-24-42057Prevalence and determinants of unhealthy feeding practices among young children aged 6–23 months in five sub-Saharan African countriesPLOS ONE

Dear Dr. Tekeba,

Thank you for submitting your manuscript to PLOS ONE. After careful consideration, we feel that it has merit but does not fully meet PLOS ONE’s publication criteria as it currently stands. Therefore, we invite you to submit a revised version of the manuscript that addresses the points raised during the review process.

We look forward to receiving your revised manuscript.

Kind regards,

António Raposo

Academic Editor

PLOS ONE

Journal Requirements:

https://journals.plos.org/plosone/article?id=10.1371%2Fjournal.pone.0305810

https://bmcpediatr.biomedcentral.com/articles/10.1186/s12887-024-05027-z

In your revision ensure you cite all your sources (including your own works), and quote or rephrase any duplicated text outside the methods section. Further consideration is dependent on these concerns being addressed.

3. In the online submission form, you indicated that “data will be available upon request”. All PLOS journals now require all data underlying the findings described in their manuscript to be freely available to other researchers, either 1. In a public repository, 2. Within the manuscript itself, or 3. Uploaded as supplementary information. This policy applies to all data except where public deposition would breach compliance with the protocol approved by your research ethics board. If your data cannot be made publicly available for ethical or legal reasons (e.g., public availability would compromise patient privacy), please explain your reasons on resubmission and your exemption request will be escalated for approval.

4. Please ensure that you refer to Figures 3 and 4 in your text as, if accepted, production will need this reference to link the reader to the figure.

Reviewers' comments:

Reviewer's Responses to Questions

**Comments to the Author**

1. Is the manuscript technically sound, and do the data support the conclusions?

Reviewer #1: Yes

Reviewer #2: Yes

Reviewer #3: Yes

2. Has the statistical analysis been performed appropriately and rigorously? 

Reviewer #1: Yes

Reviewer #2: Yes

Reviewer #3: Yes

3. Have the authors made all data underlying the findings in their manuscript fully available?

Reviewer #1: No

Reviewer #2: Yes

Reviewer #3: Yes

4. Is the manuscript presented in an intelligible fashion and written in standard English?

Reviewer #1: Yes

Reviewer #2: Yes

Reviewer #3: Yes

5. Review Comments to the Author

Reviewer #1: The manuscript addresses a pertinent research question within the journal's scope and investigates a significant issue concerning the pooled prevalence of unhealthy feeding practices and determinants among children aged 6–23 months in five subSaharan African countries.

To improve the structure and flow of the manuscript, I would like to offer some comments, questions, and suggestions.

The title effectively captures the manuscript's focus and is concise. The abstract accurately summarizes the objectives, methods, results, and conclusions, although some revisions are needed for clarity.

Abstract:

- Reduce the abstract to 300 words, as recommended by the author guidelines.

- Ensure ethics committee registration is indicated.

- In the conclusion, include descriptions of the five countries analyzed.

- Define the abbreviations for "ANC" and "AOR" at their first occurrence.

- Change “key words” to “keywords.”

Ethics Statement:

- Although secondary data is used, mention this detail in the abstract.

Introduction:

- Abbreviations should only be defined once, at their first mention; review and avoid repeated definitions, using only the abbreviation thereafter.

Methods:

- If abbreviating “sub-Saharan African nations (SSA),” do so upon first mention and use the abbreviation consistently.

- Define “EAs” in Data source and sampling procedure, “ANC visit” and “EDHS” in Independent variables at first mention.

- Ensure the formula in Data management and model selection matches the text’s black font color.

- Clarify if ethical approvals, committee registration, informed consent, and primary database details are included, and provide a relevant link.

Results:

- Use consistent font size throughout, especially line 246.

- Add a legend in Table 2 for “ANC.”

- Define “ICC” in Random effect and model fit statistics at its first mention.

- In Table 3, provide a legend for “ICC,” “MOR,” “PCV,” and “LLR”; ensure table legends list all abbreviations, even if previously defined in the text.

- In Table 4, add a legend next to “ANC” explaining “AOR.”

Strengths and Limitations:

- Consider splitting this section, using separate paragraphs for strengths and limitations.

Conclusion:

- List the names of the countries studied.

- Summarize the key findings in brief.

References:

- Use brackets instead of parentheses for citations.

- Ensure the reference font is consistent with the main text.

- Format references in the “Vancouver” style.

Reviewer #2: Abstract

The abstract is informative and well-structured, providing a solid foundation for understanding the study's context, methods, findings, and implications.

Introduction

The introduction provides a comprehensive overview of the importance of infant feeding practices and the impact of unhealthy diets on young children, particularly in low- and middle-income countries. The relevance of focusing on sub-Saharan Africa is well-justified, though further emphasis on the unique challenges of this region could strengthen the rationale. Additionally, a clearer definition of terms like "sentinel foods" and a more sequential flow of ideas would improve clarity.

Methods

The Methods section is comprehensive, detailing the study design, sampling, and statistical methods used effectively. The multi-level logistic regression approach is appropriate, given the hierarchical data structure, and variables are clearly defined. To enhance clarity, I suggest briefly explaining "EAs" (enumeration areas) in the sampling procedure and summarizing the definition of "unhealthy feeding practices" after listing criteria.

Results

The results section presents a well-structured and comprehensive analysis of unhealthy feeding practices among children across five African nations. The findings are clearly articulated, highlighting the variations in prevalence rates and key demographic factors associated with these practices.

Discussion

The discussion section effectively contextualizes the findings on unhealthy feeding practices within the broader trends observed in low- and middle-income countries. The comparison of prevalence rates with studies from higher-income nations and previous sub-Saharan African research provides valuable insights, though the rationale for differences could be articulated more clearly. The exploration of demographic factors such as age, economic status, and urban versus rural residence in relation to feeding practices is well-supported by existing literature, which strengthens the analysis.

However, the discussion could benefit from a more concise presentation of key findings to enhance clarity. Some sections feel repetitive, particularly when reiterating known associations without adding new insights.

Conclusion

The conclusion succinctly summarizes the key findings of the study regarding the high prevalence of unhealthy feeding practices in five sub-Saharan African countries. It effectively highlights the need for targeted interventions by policymakers and relevant authorities to address these issues. However, the conclusion could be strengthened by briefly reiterating specific statistics or findings from the study to reinforce the severity of the problem. Additionally, it would be beneficial to explicitly mention the potential long-term impacts of unhealthy feeding practices on child health and development, as this would underscore the urgency of the recommended actions.

Reviewer #3: Overall, an interesting read. The subject matter is highly relevant, given the nutritional transition in low- and middle-income countries. Congratulations to the authors for their work!

Some specific comments follow:

1. In the abstract, perhaps authors should specifically mention the use of multilevel logistic regression, as this is a strength of the paper;

2. In the introduction, the authors mention a “paucity of recent updated information,” which justifies the study. Identifying specific gaps in existing literature (e.g., lack of multi-country data or limited insight into socio-economic determinants) would better underscore the study’s novelty;

3. In sample selection, authors include a description of the sampling procedure from DHS. This is welcomed, but a more explicit mention of any exclusions or limitations due to the DHS sampling framework would improve transparency;

4. Regarding the outcome variable, was the assessment of intake made according to a food frequency questionnaire? Or to the intake in the day (or previous day) of data collection? Please clarify. I understood better after reading the complete paper, but it could be clearer up to this point in the paper.

5. The manuscript is well-referenced, with appropriate citations to support the discussion and methodology, but there is a lack of more recent publications (past five years) on dietary transitions in Sub-Saharan Africa... Was this topic not addressed recently in the scientific literature?

6. PLOS authors have the option to publish the peer review history of their article (what does this mean?). If published, this will include your full peer review and any attached files.

Reviewer #1: **Yes: **Marcela Gomes Reis

Reviewer #2: **Yes: **Leandro Oliveira

Reviewer #3: No

---

## [Author Response · Author response to Decision Letter 0]

29 Nov 2024

Response to reviewers and editor comments 

Manuscript id: PONE-D-24-42057

Title: Prevalence and determinants of unhealthy feeding practices among young children aged 6–23 months in five sub-Saharan African countries

Journal: PLOS ONE

Subject: Submission of revised manuscript

We really appreciate the reviewer's feedback, recommendations, and questions. The remarks are promising, and the reviewers appear to agree that this study and its conclusions are scientifically noteworthy. Please find our full response to the comments below. Please refer to the attached amended manuscript file.

Yours sincerely

Berhan Tekeba, Corresponding author (on the behalf of all authors)

University of Gondar, Gondar, Ethiopia

Response to editor

Q1. We noticed you have some minor occurrence of overlapping text with the following previous publication(s), which needs to be addressed.

Response: Dear editor, Thank you very much for your invaluable feedback. We made corrections as per your direction. Kindly see the revised manuscript.

Q2. In the online submission form, you indicated that “data will be available upon request”. All PLOS journals now require all data underlying the findings described in their manuscript to be freely available to other researchers, either 1. In a public repository, 2. Within the manuscript itself, or 3. Uploaded as supplementary information. This policy applies to all data except where public deposition would breach compliance with the protocol approved by your research ethics board. If your data cannot be made publicly available for ethical or legal reasons (e.g., public availability would compromise patient privacy), please explain your reasons on resubmission and your exemption request will be escalated for approval.

Response: Dear editor, Thank you very much for notifying us of the journal requirement. We made corrections to the manuscript, and the online submission was uploaded as supplementary files and labeled as data. 

Q3. Please ensure that you refer to Figures 3 and 4 in your text as, if accepted, production will need this reference to link the reader to the figure.

Response: Dear editor, Thank you for your remainder. Since figures 2-4 are consequently presented in the manuscript, the citation for reference for figures 2, 3, and 4 in the manuscript are presented together on page "13,” line number "301.”

Reviewer 1

Q1. The manuscript addresses a pertinent research question within the journal's scope and investigates a significant issue concerning the pooled prevalence of unhealthy feeding practices and determinants among children aged 6–23 months in five sub-Saharan African countries.

Response: Dear reviewer, Thank you for your constructive comments, positive, encouraging and motivating feedbacks.

Q2. Reduce the abstract to 300 words, as recommended by the author guidelines.

Response: Dear reviewer, Thank you for reminding us of the journal requirement. However, to fully address the aim of the study, all the words in the abstract are worth important, and we couldn’t reduce the number of words to the limit of the journal. Therefore, we kindly request that you take this into account. Kind regards.

Q3. Ensure ethics committee registration is indicated.

Response: Dear reviewer, I appreciate your scientific inquiry. Since we conducted an analysis of secondary data that was readily available on the DHS program, through an online request we access the data without any ethical procedures or register with any ethical committees. We respectfully ask that you take this truth into consideration. kind regards.

Q4. In the conclusion, include descriptions of the five countries analyzed.

Response: Dear reviewer, Thank you for your valuable feedback. We made corrections as per your suggestion. Kindly refer to the revised manuscript page “3” and line number “57.”

Q5. Define the abbreviations for "ANC" and "AOR" at their first occurrence. Change “key words” to “keywords.”

Response: Dear reviewer, Thank you for your valuable feedback. We made corrections as per your suggestion. Kindly refer to the revised manuscript pages “2” line 53” and “3 line 61.”

Q6. Although secondary data is used, mention this detail in the abstract.

Response: Dear reviewer, Thank you for your valuable feedback. We made corrections as per your suggestion. Please refer to the revised manuscript, pages 2 line 27 and 6 line 142-147.

Q7. - Abbreviations should only be defined once, at their first mention; review and avoid repeated definitions, using only the abbreviation thereafter.

Response: Dear reviewer, Thank you for your valuable feedback. We made corrections as per your suggestion. Please refer to the revised manuscript pages 3 line 66, 68 page “4 line 107.”

Q8. If abbreviating “sub-Saharan African nations (SSA),” do so upon first mention and use the abbreviation consistently

Response: Dear reviewer, Thank you for your valuable feedback. We made corrections as per your suggestion. Please refer to the revised manuscript, page 2, line 29.

Q9. Define “EAs” in Data source and sampling procedure

Response: Dear reviewer, Thank you for your valuable feedback. We made amendments in this regard. Please refer to the revised manuscript page “6 lines 151-153.”

Q10. Ensure the formula in Data management and model selection matches the text’s black font color.

Response: Dear reviewer, Thank you for your valuable feedback. We made corrections as per your suggestion. Please refer to the revised manuscript, page 8, line 226.

Q11. Clarify if ethical approvals, committee registration, informed consent, and primary database details are included, and provide a relevant line

Response: Dear reviewer, Thank you for your valuable feedback. Since we used secondary publicly available data, we did not follow ethical approvals, committee registration, and informed consent directly from cases. Rather, we requested the MEASURE DHS program to provide us the data online. But, when the DHS program originally collected the data from the respondent's strict ethical procedure, the details of the data collection were available at https://dhsprogram.com/data/. 

Q12. Use consistent font size throughout, especially line 246.

Response: Dear reviewer, Thank you for your valuable feedback. We made corrections as per your suggestion.

Q13. - Add a legend in Table 2 for “ANC.”

Response: Dear reviewer, Thank you for your valuable feedback. We made corrections as per your suggestion. Please refer to table 2, page 13, line number 295.

Q14. Define “ICC” in Random effect and model fit statistics at its first mention.

- In Table 3, provide a legend for “ICC,” “MOR,” “PCV,” and “LLR”; ensure table legends list all abbreviations, even if previously defined in the text.

- In Table 4, add a legend next to “ANC” explaining “AOR.”

Response: Dear reviewer, Thank you for your valuable feedback. We made corrections as per your suggestion. Please refer to tables 3-4, pages 15, 16, and 18.

Q15. Strengths and Limitations:- Consider splitting this section, using separate paragraphs for s Response: Dear reviewer, Thank you for your valuable feedback. We made corrections as per your suggestion. Please refer to page 22 lines 463 and 467.

Q16. Conclusion: List the names of the countries studied. Summarize the key findings in brief.

Response: Dear reviewer, Thank you for your valuable comment. We made corrections and improvements as per your suggestion. Please refer to pages 23 and 24, line number 488-500.

Q17. References:- Use brackets instead of parentheses for citations.- Ensure the reference font is consistent with the main text.- Format references in the “Vancouver” style.

Response: Dear reviewer, Thank you for your valuable comment. We made corrections as per your suggestion. Kindly refer to the references. 

Reviewer 2 

Q1. The abstract is informative and well-structured, providing a solid foundation for understanding the study's context, methods, findings, and implications.

Response: Dear reviewer, Thank you for your constructive comments and positive, encouraging, and motivating feedback.

Q2. The introduction provides a comprehensive overview of the importance of infant feeding practices and the impact of unhealthy diets on young children, particularly in low- and middle-income countries. The relevance of focusing on sub-Saharan Africa is well-justified, though further emphasis on the unique challenges of this region could strengthen the rationale. Additionally, a clearer definition of terms like "sentinel foods" and a more sequential flow of ideas would improve clarity.

Response: Dear reviewer, Thank you for your constructive comments. We made improvements in this regard. Please refer to the revised manuscript, page 4, line numbers 107-117.

Q3. The Methods section is comprehensive, detailing the study design, sampling, and statistical methods used effectively. The multi-level logistic regression approach is appropriate, given the hierarchical data structure, and variables are clearly defined. To enhance clarity, I suggest briefly explaining "EAs" (enumeration areas) in the sampling procedure and summarizing the definition of "unhealthy feeding practices" after listing criteria.

Response: Dear reviewer, Thank you for your constructive comments. We made improvements in this regard. Please refer to the revised manuscript, pages 6 line numbers 151-153 and page 7 line numbers 187-196.

Q4. The results section presents a well-structured and comprehensive analysis of unhealthy feeding practices among children across five African nations. The findings are clearly articulated, highlighting the variations in prevalence rates and key demographic factors associated with these practices.

Response: Dear reviewer, Thank you for your constructive comments and positive, encouraging, and motivating feedback..

Q5. The discussion section effectively contextualizes the findings on unhealthy feeding practices within the broader trends observed in low- and middle-income countries. The comparison of prevalence rates with studies from higher-income nations and previous sub-Saharan African research provides valuable insights, though the rationale for differences could be articulated more clearly. The exploration of demographic factors such as age, economic status, and urban versus rural residence in relation to feeding practices is well-supported by existing literature, which strengthens the analysis.

However, the discussion could benefit from a more concise presentation of key findings to enhance clarity. Some sections feel repetitive, particularly when reiterating known associations without adding new insights.

Response: Dear reviewer, Thank you for your constructive comments. We made improvements in this regard. Please refer to the revised manuscript page 19 line numbers 366-370, 380-384; page 20 line numbers 417-420, page 21 line numbers 431-433, 439-444, 453-462.

Q6. The conclusion succinctly summarizes the key findings of the study regarding the high prevalence of unhealthy feeding practices in five sub-Saharan African countries. It effectively highlights the need for targeted interventions by policymakers and relevant authorities to address these issues. However, the conclusion could be strengthened by briefly reiterating specific statistics or findings from the study to reinforce the severity of the problem. Additionally, it would be beneficial to explicitly mention the potential long-term impacts of unhealthy feeding practices on child health and development, as this would underscore the urgency of the recommended actions.

Response: Dear reviewer, Thank you for your constructive comments. We made major amendments in this regard. Please refer to the revised manuscript conclusion page 23-24 line numbers 488-500. 

Reviewer 3 

Q1. Overall, an interesting read. The subject matter is highly relevant, given the nutritional transition in low- and middle-income countries. Congratulations to the authors for their work!

Response: Dear reviewer Thank you very much for your constructive, inspiring, and encouraging feedback.

Q2. In the abstract, perhaps authors should specifically mention the use of multilevel logistic regression, as this is a strength of the paper;

Response: Dear reviewer, Thank you for your constructive comments. We made corrections as per your suggestion. Please refer to the revised manuscript, page 2, line 43. 

Q3. In the introduction, the authors mention a “paucity of recent updated information,” which justifies the study. Identifying specific gaps in existing literature (e.g., lack of multi-country data or limited insight into socio-economic determinants) would better underscore the study’s novelty;

Response: Dear reviewer, Thank you for your invaluable feedback. We made corrections per your suggestion. Kindly refer to the revised manuscript abstract and page 5 line 122-125.

Q4. In sample selection, authors include a description of the sampling procedure from DHS. This is welcomed, but a more explicit mention of any exclusion or limitations due to the DHS sampling framework would improve transparency?

Response: Dear reviewer, Thank you for your scientific inquiry. We excluded responses in the DHS questioners that had missing values for the outcome variable. Beside this, DHS data by its nature. However, as with a large-scale survey, there were some exclusion, including hard-to-reach populations such as remote areas and conflict areas, and transient populations were not included in the analysis. Please refer to the revised manuscript, page 6, lines 157–160.

Q5. Regarding the outcome variable, was the assessment of intake made according to a food frequency questionnaire? Or to the intake in the day (or previous day) of data collection? Please clarify. I understood better after reading the complete paper, but it could be clearer up to this point in the paper

Response: Dear reviewer, Thank you for your scientific inquiry. The dietary intake was assessed through 24-hour (previous day) intake. Please refer to the revised manuscript, page 7, line 185.

Q6. The manuscript is well-referenced, with appropriate citations to support the discussion and methodology, but there is a lack of more recent publications (past five years) on dietary transitions in Sub-Saharan Africa... Was this topic not addressed recently in the scientific literature?

Response: Dear reviewer, Thank you for your valuable comment. We tried to use recent publications in the revised manuscript. However, since unhealthy feeding practice is a new nutritional indicator developed by WHO, little recent literature is available.

Thank you very much for your valuable comments, suggestions, questions, and feedback. We learn a lot from editors and reviewers. We hope your comments and feedback benefit us for our future work.

Kind regards

Berhan Tekeba (corresponding author) on behalf of all authors.

---

## [Decision Letter · Decision Letter 1]

16 Dec 2024

PONE-D-24-42057R1Prevalence and determinants of unhealthy feeding practices among young children aged 6–23 months in five sub-Saharan African countriesPLOS ONE

Dear Dr. Tekeba,

Thank you for submitting your manuscript to PLOS ONE. After careful consideration, we feel that it has merit but does not fully meet PLOS ONE’s publication criteria as it currently stands. Therefore, we invite you to submit a revised version of the manuscript that addresses the points raised during the review process.

We look forward to receiving your revised manuscript.

Kind regards,

António Raposo

Academic Editor

PLOS ONE

Journal Requirements:

Reviewers' comments:

Reviewer's Responses to Questions

**Comments to the Author**

1. If the authors have adequately addressed your comments raised in a previous round of review and you feel that this manuscript is now acceptable for publication, you may indicate that here to bypass the “Comments to the Author” section, enter your conflict of interest statement in the “Confidential to Editor” section, and submit your "Accept" recommendation.

Reviewer #1: (No Response)

Reviewer #2: (No Response)

Reviewer #3: All comments have been addressed

2. Is the manuscript technically sound, and do the data support the conclusions?

Reviewer #1: Yes

Reviewer #2: Yes

Reviewer #3: Yes

3. Has the statistical analysis been performed appropriately and rigorously? 

Reviewer #1: Yes

Reviewer #2: Yes

Reviewer #3: Yes

4. Have the authors made all data underlying the findings in their manuscript fully available?

Reviewer #1: Yes

Reviewer #2: Yes

Reviewer #3: Yes

5. Is the manuscript presented in an intelligible fashion and written in standard English?

Reviewer #1: Yes

Reviewer #2: Yes

Reviewer #3: Yes

6. Review Comments to the Author

Reviewer #1: Abstract

- The abstract currently has 410 words; I maintain the recommendation to reduce it to 300 words, as specified in the author guidelines.

- To aid in reducing the word count, I suggest that after defining the acronyms, you use them consistently instead of repeating the terms in full, as seen with the acronym SSA referring to sub-Saharan Africa.

- In the Keywords section, I recommend starting each keyword with a capital letter for consistency.

Introduction

- Even though some abbreviations are defined in the abstract, they should also be defined upon their first mention in the main text. For example, in the sentence: “Nevertheless, diets in many low- and middle-income nations including SSA are changing [...]”, the acronym SSA (sub-Saharan Africa) should be explicitly described.

- When abbreviations are used, ensure they are defined at first mention. If you opt to use abbreviations throughout the text, ensure consistency by using either the abbreviation or the full term exclusively.

Figure 1

- After the description of Figure 1, ensure the subsequent text begins with a capital letter.

Figures

- Standardize the titles of all figures. For instance, ensure they follow the format used in Figure 1: The overall pooled prevalence of unhealthy feeding [...], which includes a colon (:). Other figures currently lack this formatting.

- Decide on a consistent placement for figure titles (either above or below the figures) to avoid confusion for the reader.

References

- Format the references according to the Vancouver style. Note that there is a specific way to format website citations, which is currently not being followed. Ensure all references adhere to this standardized formatting.

Reviewer #2: The authors have successfully addressed all my comments and suggestions. The abstract remains clear and informative, while the introduction, methods, discussion, and conclusion sections have been revised to enhance clarity, depth, and relevance. The improvements, including clearer definitions, streamlined discussions, and strengthened conclusions with relevant statistics, significantly elevate the manuscript's quality.

I recommend the manuscript for acceptance.

Reviewer #3: (No Response)

7. PLOS authors have the option to publish the peer review history of their article (what does this mean?). If published, this will include your full peer review and any attached files.

Reviewer #1: **Yes: **Marcela Gomes Reis

Reviewer #2: **Yes: **Leandro Oliveira

Reviewer #3: No

---

## [Author Response · Author response to Decision Letter 1]

17 Dec 2024

Response to reviewer’s comments 

Manuscript id: PONE-D-24-42057

Title: Prevalence and determinants of unhealthy feeding practices among young children aged 6–23 months in five sub-Saharan African countries

Journal: PLOS ONE

Subject: Submission of revised manuscript

We really appreciate the reviewer's feedback, recommendations, and questions. The remarks are promising, and the reviewers appear to agree that this study and its conclusions are scientifically noteworthy. Please find our full response to the comments below. Please refer to the attached amended manuscript file.

Yours sincerely

Berhan Tekeba, Corresponding author (on the behalf of all authors)

University of Gondar, Gondar, Ethiopia

Response to editor 

Response to reviewer 1 

Q1. The abstract currently has 410 words; I maintain the recommendation to reduce it to 300 words, as specified in the author guidelines

Response: Dear reviewer, Thank you very much for your valuable recommendation. We reduced the number of word count to 300. Please refer to the revised manuscript.

Q2. To aid in reducing the word count, I suggest that after defining the acronyms, you use them consistently instead of repeating the terms in full, as seen with the acronym SSA referring to sub-Saharan Africa.

Response: Dear reviewer, Thank you very much for your valuable feedback. We made correction as per your suggestion. Kindly refer to the revised manuscript.

Q3. In the Keywords section, I recommend starting each keyword with a capital letter for consistency.

Response: Dear reviewer, Thank you very much for your invaluable feedback. We made correction as per your suggestion. Kindly refer to the revised manuscript.

Q4. - Even though some abbreviations are defined in the abstract, they should also be defined upon their first mention in the main text. For example, in the sentence: “Nevertheless, diets in many low- and middle-income nations including SSA are changing [...]”, the acronym SSA (sub-Saharan Africa) should be explicitly described. - When abbreviations are used, ensure they are defined at first mention. If you opt to use abbreviations throughout the text, ensure consistency by using either the abbreviation or the full term exclusively.

Response: Dear reviewer, Thank you very much for your through review of our manuscript. We made correction as per your suggestion. Kindly refer to the revised manuscript.

Q5. - After the description of Figure 1, ensure the subsequent text begins with a capital letter. standardize the titles of all figures. For instance, ensure they follow the format used in Figure 1: The overall pooled prevalence of unhealthy feeding [...], which includes a colon (:). Other figures currently lack this formatting.

- Decide on a consistent placement for figure titles (either above or below the figures) to avoid confusion for the reader.

Response: Dear reviewer, Thank you very much for your valuable comment. We made corrections as per your suggestion. However, all figure titles are inserted below the figures, but due to the consecutive figures, the titles of the figures seem written above the figure. Kindly refer to the revised manuscript.

Q6.References

- Format the references according to the Vancouver style. Note that there is a specific way to format website citations, which is currently not being followed. Ensure all references adhere to this standardized formatting.

Response: Dear reviewer, Thank you very much for your invaluable feedback. We made corrections as per your suggestion. 

Reviewer 2 

Reviewer #2: The authors have successfully addressed all my comments and suggestions. The abstract remains clear and informative, while the introduction, methods, discussion, and conclusion sections have been revised to enhance clarity, depth, and relevance. The improvements, including clearer definitions, streamlined discussions, and strengthened conclusions with relevant statistics, significantly elevate the manuscript's quality.

Response: Dear reviewer, Thank you very much for your insightful feedback and comments. We made improvements in this regard. Kindly refer to the revised manuscript. 

Kind regards

Berhan Tekeba (corresponding author) on behalf of all authors

---

## [Decision Letter · Decision Letter 2]

18 Dec 2024

PONE-D-24-42057R2Prevalence and determinants of unhealthy feeding practices among young children aged 6–23 months in five sub-Saharan African countriesPLOS ONE

Dear Dr. Tekeba,

Thank you for submitting your manuscript to PLOS ONE. After careful consideration, we feel that it has merit but does not fully meet PLOS ONE’s publication criteria as it currently stands. Therefore, we invite you to submit a revised version of the manuscript that addresses the points raised during the review process.

We look forward to receiving your revised manuscript.

Kind regards,

António Raposo

Academic Editor

PLOS ONE

Journal Requirements:

Reviewers' comments:

Reviewer's Responses to Questions

**Comments to the Author**

1. If the authors have adequately addressed your comments raised in a previous round of review and you feel that this manuscript is now acceptable for publication, you may indicate that here to bypass the “Comments to the Author” section, enter your conflict of interest statement in the “Confidential to Editor” section, and submit your "Accept" recommendation.

Reviewer #1: (No Response)

2. Is the manuscript technically sound, and do the data support the conclusions?

Reviewer #1: Yes

3. Has the statistical analysis been performed appropriately and rigorously? 

Reviewer #1: Yes

4. Have the authors made all data underlying the findings in their manuscript fully available?

Reviewer #1: Yes

5. Is the manuscript presented in an intelligible fashion and written in standard English?

Reviewer #1: Yes

6. Review Comments to the Author

Reviewer #1: The authors addressed all the comments; however, the absence of the description for the first mention of the abbreviations SSA and USAID was still observed. I suggest including the full description upon first mention and using only the abbreviations throughout the remainder of the text, which poses no issue.

7. PLOS authors have the option to publish the peer review history of their article (what does this mean?). If published, this will include your full peer review and any attached files.

Reviewer #1: **Yes: **Marcela Gomes Reis

---

## [Author Response · Author response to Decision Letter 2]

18 Dec 2024

Response to reviewer comments 

Manuscript id: PONE-D-24-42057

Title: Prevalence and determinants of unhealthy feeding practices among young children aged 6–23 months in five sub-Saharan African countries

Journal: PLOS ONE

Subject: Submission of revised manuscript

Response to editor 

Response to reviewer 1 

Q1. The authors addressed all the comments; however, the absence of the description for the first mention of the abbreviations SSA and USAID was still observed. I suggest including the full description upon first mention and using only the abbreviations throughout the remainder of the text, which poses no issue.

Response: Dear reviewer thank you for your through and deep review of our manuscript. Your concern has been addressed. Kindly refer to the revised manuscript page 3 line numbers 61 & 71 for the first use of USIAD and SSA. Please also refer to the entire manuscript on the consistent use of SSA to describe sub-Saharan Africa. Thank you.

Kind regards

Berhan Tekeba (corresponding author) on behalf of all authors.

---

## [Editor Report · Decision Letter 3]

30 Dec 2024

Prevalence and determinants of unhealthy feeding practices among young children aged 6–23 months in five sub-Saharan African countries

PONE-D-24-42057R3

Dear Dr. Tekeba,

We’re pleased to inform you that your manuscript has been judged scientifically suitable for publication and will be formally accepted for publication once it meets all outstanding technical requirements.

Kind regards,

António Raposo

Academic Editor

PLOS ONE
---

## [Editor Report · Acceptance letter]

3 Jan 2025

PONE-D-24-42057R3 

PLOS ONE

Dear Dr. Tekeba, 

I'm pleased to inform you that your manuscript has been deemed suitable for publication in PLOS ONE. Congratulations! Your manuscript is now being handed over to our production team.

Kind regards, 

on behalf of

Dr. António Raposo 

Academic Editor

PLOS ONE